# Merkel Cell Polyomavirus in Merkel Cell Carcinoma: Integration Sites and Involvement of the KMT2D Tumor Suppressor Gene

**DOI:** 10.3390/v12090966

**Published:** 2020-08-31

**Authors:** Reety Arora, Jae Eun Choi, Paul W. Harms, Pratik Chandrani

**Affiliations:** 1Cellular Organization and Signalling Group, National Centre for Biological Sciences, Tata Institute of Fundamental Research, Bangalore 560065, India; 2Department of Pathology, University of Michigan, Ann Arbor, MI 48109, USA; jaeeumc@med.umich.edu (J.E.C.); paulharm@med.umich.edu (P.W.H.); 3Michigan Center for Translational Pathology, University of Michigan, Ann Arbor, MI 48109, USA; 4Rogel Cancer Center, University of Michigan, Ann Arbor, MI 48109, USA; 5School of Medicine, University of San Diego, San Diego, CA 92093, USA; 6Department of Dermatology, University of Michigan, Ann Arbor, MI 48109, USA; 7Medical Oncology Molecular Laboratory, Medical Oncology Department, Tata Memorial Hospital, Mumbai 400012, India; pratikchandrani@gmail.com; 8Centre for Computational Biology, Bioinformatics and Crosstalk Laboratory, ACTREC–Tata Memorial Centre, Navi Mumbai 410210, India

**Keywords:** Merkel cell polyomavirus, Merkel cell carcinoma, viral integration, KMT2D

## Abstract

Merkel cell carcinoma (MCC) is an uncommon, lethal cancer of the skin caused by either Merkel cell polyomavirus (MCPyV) or UV-linked mutations. MCPyV is found integrated into MCC tumor genomes, accompanied by truncation mutations that render the MCPyV large T antigen replication incompetent. We used the open access HPV Detector/Cancer-virus Detector tool to determine MCPyV integration sites in whole-exome sequencing data from five MCC cases, thereby adding to the limited published MCPyV integration site junction data. We also systematically reviewed published data on integration for MCPyV in the human genome, presenting a collation of 123 MCC cases and their linked chromosomal sites. We confirmed that there were no highly recurrent specific sites of integration. We found that chromosome 5 was most frequently involved in MCPyV integration and that integration sites were significantly enriched for genes with binding sites for oncogenic transcription factors such as LEF1 and ZEB1, suggesting the possibility of increased open chromatin in these gene sets. Additionally, in one case we found, for the first time, integration involving the tumor suppressor gene *KMT2D*, adding to previous reports of rare MCPyV integration into host tumor suppressor genes in MCC.

## 1. Introduction

Merkel cell carcinoma (MCC) is an aggressive skin cancer (five year overall survival rate of about 44% and disease-specific mortality of 33–46% [1,2]), predominantly arising in the elderly and immunocompromised. MCC is associated with Merkel cell polyomavirus (MCPyV) in the majority of cases [2,3]. MCPyV was discovered in 2008 by Feng et al. and found to be integrated into the MCC tumor genome in a mutated, replication-deficient form [3]. MCPyV is a small (5 kb), non-enveloped, circular dsDNA polyomavirus that is commonly found on human skin [2,3,4]. The MCPyV early region expresses large T antigen (LT) and small T antigen (sT) proteins that have been shown to drive tumorigenesis and are likely causative for MCC [5,6]. MCC tumor cells express a truncated form of LT protein that cannot mediate viral replication, but it retains the domain responsible for inhibition of the tumor suppressor Rb [2,3,4,5]. MCC that lack MCPyV display cellular genomic mutations in tumor suppressor genes, especially *TP53* and *RB1* [7,8,9]. Additional tumor suppressors such as *NOTCH* family genes and *KMT2D* are inactivated at lower rates in MCPyV-negative MCC [2,7,8,9].

Integration into the human genome is implicated as an early oncogenic event for many DNA tumor viruses [10,11]. Integration increases the risk of cancer beyond simple viral infection as it can disrupt and deregulate nearby human cancer-associated genes, contribute to genome instability, and create fusion transcripts [11,12]. Various studies show an overall bias for integration near open chromatin regions and SINE (Short Interspersed Nuclear Elements) elements for DNA tumor viruses [12]. Interestingly, there are differences between integration site patterns related to different virus types, cancer types, and disease stages.

Because MCPyV infection is near-ubiquitous [10], demonstrations of viral integration can be helpful in distinguishing tumor-associated viral sequences from spurious detection of background viruses. Furthermore, viral integration can interrupt tumor suppressor genes, representing a potential mechanism for viral tumorigenesis in addition to viral oncogenes. Despite the diagnostic and biologic importance of this question, studies investigating MCPyV integration sites have been relatively few due to the rarity of MCC and limited high-quality tumor samples. Furthermore, the identification of MCPyV integration is technically challenging as the circular genome of the virus does not linearize in a consistent manner. Several approaches have been developed to address this question, often relying upon custom assays or analysis tools (Table 1 and Appendix A). A sensitive approach for detecting MCPyV integration sites using commercially available assays and open-access analysis tools has not yet been uniformly validated and adopted.

MCPyV integration sites have been reported across most chromosomes and may occur within genes or in intergenic regions (Table 2). With rare exceptions, the integration site for a particular tumor (and its related metastases) is unique to that case. Despite this variability, several studies have suggested that the distribution of integration sites is non-random, with patterns related to mechanisms of integration [8,9,13,14,15]. Unlike HIV, MCPyV does not carry an integrase, and integration into the host genome is not part of the viral life cycle [3]. Starrett et al. reported that integration is likely mediated through erroneous DNA repair at sites of microhomology between the host and viral genomes, similar to mechanisms identified for HPV genome integration in squamous cell carcinoma tumors [8,9]. Similarly, Czech-Sioli et al. recently observed that MCPyV follows two main types of integration site structure: (1) linear patterns with chromosomal breakpoints related to non-homologous end joining; and (2) complex integration loci dependent on microhomology-mediated break-induced replication [16,17]. Doolittle-Hall et al. conducted the first meta-analysis on 37 MCCs for integration analysis and found that MCPyV integration occurred preferentially near SINEs and BDP1 binding sites [12]. They also confirmed viral preference for transcriptionally active gene-dense regions and accessible chromatin. In addition, they observed a tropism toward integration near specific categories of genes, including sensory perception and G-protein coupled receptor genes [12]. MCPyV integration within cellular genes could also act as a mechanism of tumor suppressor gene inactivation in a subset of tumors, as potentially disruptive integration into genes with tumor suppressor roles (*PTPRG*, *XRCC4*) has been described in individual cases. However, recurrent disruption of specific tumor suppressors has not been described [3,8,9,12,13,14,15,16,18,19,20,21] (Appendix A and Table 2).

Although most tumors demonstrate a single, unique genomic integration site, rare exceptions have been described. Starrett et al., 2020, found that two independent tumors had overlapping (although non-identical) integration sites on chromosome 1, representing a rare example of a recurrent integration site. An additional case had three distinct integration sites on different chromosomes within a single tumor: one full copy of the viral genome, one smaller segment predicted to retain oncogenic activity (containing NCCR and T antigen coding regions), and one smaller segment predicted to lack oncogenic activity [9].

Patterns by which the circular MCPyV viral genome linearizes and integrates have been defined by several studies (Table 1) [8,9]. For both MCPyV- and HPV-mediated tumors, integration may be followed by the formation of a transiently circular DNA intermediate containing the viral genome and flanking regions of the host genome which can be further amplified through aberrant firing of the viral origin of replication. This piece then reintegrates into the host genome, appearing as amplified regions of the host genome in a tandem head-to-tail conformation interspersed with the viral genome [8,9].

Although the breakpoint in the MCPyV genome is highly variable, most studies describe that junctions are predominantly located in exon 2 of the LT gene after the pRb binding site (Table 2). This could result in a truncated, replication-deficient LT protein equivalent to tumor-specific LT truncating mutations.

More comprehensive characterization of MCPyV integration locations, and thereby preferences, will advance the understanding of virus-mediated tumorigenesis and may reveal therapeutic vulnerabilities. Our study aimed at reviewing published MCPyV integration site data and expanding this repertoire of known MCPyV integration sites. We adapted a bioinformatics tool designed for HPV integration site detection (HPV Detector/Cancer-virus Detector [22]) to detect MCPyV from previously described MCC whole-exome sequencing datasets from patient tumor samples classified as MCPyV-positive by quantitative PCR [7]. Our results identify additional specific viral and host junction sites, including integration into a tumor suppressor gene frequently mutated in MCPyV-negative MCC (*KMT2D*). Furthermore, we demonstrate the successful use of HPV Detector/Cancer-virus Detector, a simple open-access analysis tool with potential utility in research and clinical testing, for detection of MCPyV integration in standard whole-exome sequencing datasets.

## 2. Methods 

### 2.1. MCPyV Integration Analysis

MCC whole-exome datasets were taken from a previously published study by Harms et al., Cancer Research, 2015 [7]. The sequencing approach used by this study is briefly summarized here. Exome libraries were generated using an Illumina TruSeq DNA Sample Prep Kit, and paired-end libraries were sequenced with the Illumina HiSeq 2000. Reads that passed the Illumina BaseCall software chastity filter were used for subsequent analysis. Sequence and alignment quality were assessed by FastQC and the Picard package, respectively. More details can be found in the supplement of Harms et al., Cancer Research, 2015 [7].

The HPV Detector used a computational subtraction approach to identify HPV DNA traces from NGS datasets, as published earlier [22]. We modified the HPV Detector to align NGS data to human and MCPyV genomes. In the downstream analysis, we subtracted all NGS reads that were mapped to only the human genome. The remaining reads were either mapped to the MCPyV genome completely or were split reads with parts mapped to the human genome and other parts mapped to the MCPyV genome. We identified all split reads and annotated them using nearby human and MCPyV genome regions through BEDTools [23]. The source code can be accessed via GitHub (https://github.com/pratikchandrani). The sequencing datasets generated previously in Harms et al., 2015 [7], were used for this study. The integration site junctions/genes were further subjected to pathway analysis using GSEA-MSigDB [24] under the default parameters.

### 2.2. PCR

Primers were designed at the flanking regions (one primer on the human side and another on the viral side) to amplify the integration breakpoints. PCR was performed on genomic DNA from the MCC tumor samples using Platinum Taq DNA Polymerase (catalog # 10966018, Invitrogen, MI, USA), DMSO spike, and 20ng of DNA. Water was used as a negative control.

### 2.3. MCPyV Integration Site Meta-Analysis

Based on the closest chromosome band for each integration junction on the human genome, the coordinates were taken from UCSC Browser, GRCh38/hg38. The chromosome and start positions were then plotted by Circo plot to represent the integration sites. COSMIC v91 data were downloaded from the Sanger Institute website (https://cancer.sanger.ac.uk/census) [25], and the Human Fragile Sites dataset was downloaded from the HumCFS database (https://webs.iiitd.edu.in/raghava/humcfs/) [26]. The Circo plot was drawn using Circa Software (OMGenomics http://omgenomics.com/circa/). Of the 123 sites for which the chromosome number was available, 96 (including five from our study) reported information regarding location on the chromosome. Using chromosome band information, a Circo plot was drawn for these 96 sites. Dots on the bars represent the location of genes in which integration sites were found.

## 3. Result

### 3.1. Integration Site Detection from Exome Sequencing Data

Of the seven MCC whole-exome sequencing datasets previously published by Harms et al. [7], we predicted viral–host junctions in five (including one primary tumor and four metastases). Similar to previous studies, we found that the integration site was highly variable and at either interchromosomal locations or in introns [8,12,20]. Three of these five sites were found in intergenic regions in chromosomes 4, 7, and 16, whereas for two tumors the sites were found in introns of two genes: *KMT2D* (intron 34) on chromosome 12 and *DCAF7* (intron 3) on chromosome 17 (Table 1, Table 3 and Figure 1A). *KMT2D* is a lysine methyltransferase and tumor suppressor gene. *DCAF7* (DDB1 and CUL4 Associated Factor 7) encodes a scaffold protein for kinase signaling. Unlike findings from a previous report [12], we did not observe a trend toward involvement of sensory genes.

Recent sequencing studies have shown that chromatin remodeling and histone modulators are frequently altered in cancers, with implications for tumor biology. KMT2D (lysine (K)-specific methyltransferase 2D), also known as MLL2 (myeloid/lymphoid or mixed-lineage leukemia 2, also known as ALR/MLL4), is one such histone methyltransferase that plays an important role in regulating gene transcription [27]. In particular, it targets lysine 4 of histone H3 (H3K4) whose methylations serve as a gene activation mark. KMT2D is essential for maintaining the level of global H3K4 monomethylation, with its enzymatic SET domain being directly responsible for this function. Furthermore, most KMT2D binding sites are located in regions of potential enhancer elements. KMT2D has [27] emerged as a frequently mutated gene in about 8.5% of all cancers [28] and has been proposed to play either a tumor suppressor or oncogenic role, depending on cancer type. In MCC, KMT2D predominantly demonstrates inactivating mutations, thus supporting its tumor suppressor role [29].

To determine the integration junction for the viral genome, we mapped the sequences to MCPyV DysKA isolate [30] (GenBank ID # KX781279.1). The sites we found differed from the previously reported pattern in which integration junctions predominantly involved the LT 2nd exon. In our cases we found two integration sites in the NCCR region, two in the VP1 region, and only one in LT 2nd exon (Table 1, Table 3 and Figure 1B). Intriguingly, the sequence we found for LT at the junction is preceded by a TAA sequence in the MCPyV genome, indicating that this integration could also be associated with the generation of a premature LT stop codon, similar to findings by Schrama et al. [18].

Several cases showed integration sites in the NCCR region (of these, our study reports two of the total eight), VP1 (our study reports two of the total 23), and VP2 regions (total of 10 reported). The integration junctions may be disruptive or non-disruptive in gene expression and function, depending on the frame of fused product, mechanism of the integration, number of viral copies integrated, etc. Based upon the previous description of our cases, six of seven tumors had detectable LT protein expression by immunohistochemistry, and six of six tumors with RNASeq data had detectable LT transcript expression (Harms et al., 2015, Supplemental Material) [7]. Therefore, all integration sites were permissive for LT expression, even those involving the NCCR. Of note, our approach is not designed to evaluate effects of integration on LT (especially relevant for integration sites that truncate the protein).

By PCR and using primers flanking the virus–host junctions, followed by Sanger sequencing of the resulting PCR amplicon, we validated the integration sites predicted for three of these cases (Figure 1C). The remaining cases lacked adequate remaining DNA for PCR.

### 3.2. Correlation with Previously Reported MCPyV Integration Sites

We evaluated our 5 integration site results in the context of 91 other previously reported MCPyV integration sites into the human genome by using Circo plot [31] overlaid with COSMIC gene datasets and fragile sites [26]. Chromosome 5 stands out with the highest (21) MCC cases having integrated MCPyV. There are no integration sites in chromosomes 21, 22, and X. Only five genes of the many associated with MCPyV integration, namely *SF3B1, TLX3, CD74*, *MYC,* and *KMT2D,* are cancer-linked genes listed in the COSMIC database [25]. Additional genes with potential tumor suppressor function that have been shown to be involved in MCPyV integration include *PTPRG*, *GRWD1*, and *XRCC4* [3,19,21]. Hence, in a minority of MCPyV-positive MCC tumors, MCPyV integration itself may contribute to oncogenesis in a variable and case-specific manner via disruption of cancer genes. Additionally, although there is some overlay with fragile sites as previously reported, this effect did not appear significant. Furthermore, the integration junctions in the human genome were observed to be enriched 91in genes having motifs/binding sites of known oncogenic transcription factors such as LEF1 [32], MYB [33], and AREB6 (ZEB1) [34] along with other pathways (Appendix A). We speculate this integration pattern might be related to an increased probability of open chromatin for these gene sets (Figure 2).

For the integration junctions in the viral genome (89 in total including this study), we observed the previously reported bias to the T antigen region, with 42 cases displaying junctions in LT, and 4 in sT (Table 2).

### 3.3. Limitations

We were limited by the amount and quality of available MCC tumor DNA for PCR-Sanger validation, as the majority was used for next generation sequencing. Similarly, previously described inverse PCR approaches for demonstrating viral genome concatemerization [18] were not possible due to limited material. In addition, our approach relies upon identification of off-target capture or shoulder reads of viral sequences within human whole-exome sequencing data, thus generating relatively less sequencing depth than a sequencing approach specifically targeting MCPyV. We also cannot exclude the possibility that the lack of detected integration sites in 2/7 tumors may represent a technical limitation of our approach rather than true absence of viral integration.

## 4. Conclusions

Our analysis adds to the repertoire of MCPyV integration junction data and systematically collates current information regarding the same. We show unique genetic sites of integration for both the virus and host, confirming the highly variable nature of MCPyV integration. For the first time, we demonstrate integration involving the tumor suppressor gene *KMT2D*, representing the potential inactivation of a tumor suppressor gene also recurrently inactivated in MCPyV-negative MCC. We also demonstrate that data from tumor whole-exome sequencing data can be effectively interrogated by the simple and open-access Cancervirus Detector tool to demonstrate viral integration sites, thus representing a powerful and accessible new approach for analyzing existing sequencing datasets.

## Figures and Tables

**Figure 1 viruses-12-00966-f001:**
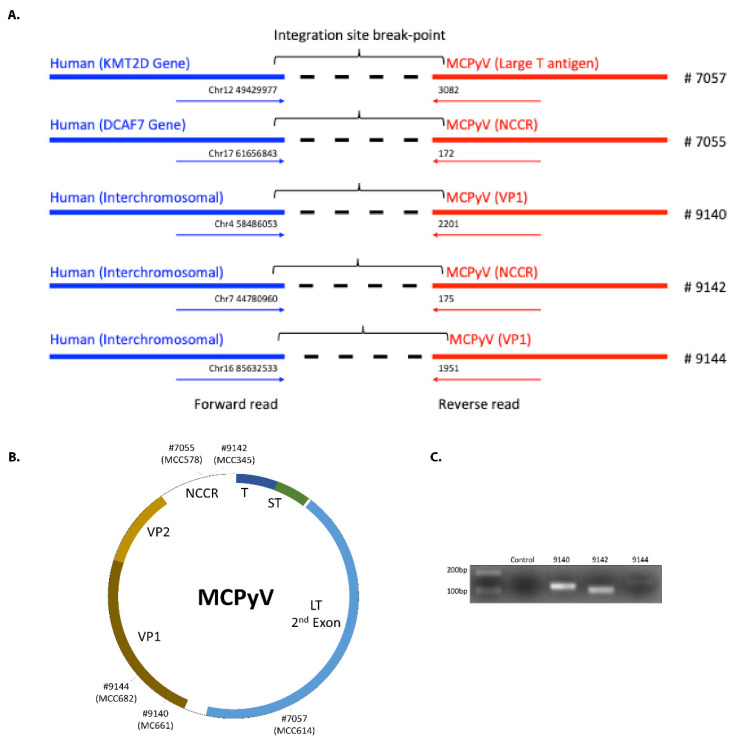
(**A**). Integration breakpoints for the different Merkel cell carcinoma (MCC) cases. Blue is the human sequence detected and red is the MCPyV sequence. (**B**) Integration junctions identified for samples from the current study, by sample number and original tumor ID. (**C**) PCR validation of integration break points. LT: large T antigen, 2nd exon. NCCR: non-coding control region. ST: small T antigen. T: common T antigen exon. VP: viral protein (capsid).

**Figure 2 viruses-12-00966-f002:**
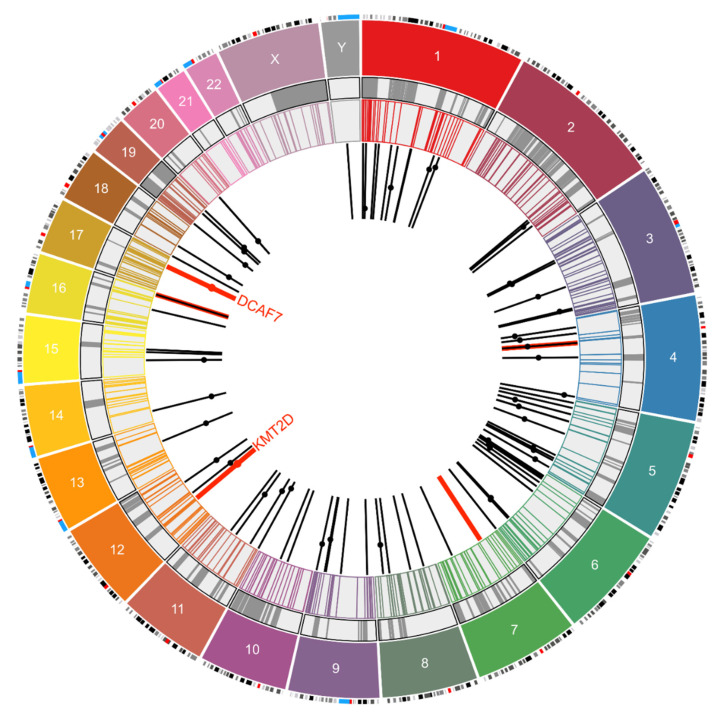
Circo plot representing 96 integration sites of MCPyV mapped to different chromosomes. The outermost layers represent the chromosomes (color coded) and their respective chromosome bands. The grey circle marks the human fragile sites from the HumCFS database, and the next concentric circle marks the COSMIC database gene sites. The black bars mark 91 integration locations of MCPYV in MCC from previous reports. Dots on the bars represent the human genes associated with this integration event, but their position on the bar is arbitrary. The red bars and dots are data from five MCC cases analyzed using the HPV Detector in this study. The Circo plot was drawn using Circa Software (OMGenomics http://omgenomics.com/circa/).

**Table 1 viruses-12-00966-t001:** Summary of studies on MCPyV integration in the human genome.

Year	First Author (Journal)	Detection Assay	Integration Analysis Tool	Number of MCC—Integration Sites Detected	Human Genome Integration Junction, Chromosome (# of Cases)	MCPyV Genome Integration Junction (# of Cases)
2008	Feng H. (Science)	cDNA pyrosequencing, 3’-RACE, 5’-RACE, Southern Hybridization	Digital Transcriptome Subtraction	2	chr3	LT (1)
2009	Sastre-Garau X. (J Pathol)	DIPS-PCR	N/A	10	chr2, 3, 4, 5, 6, 8, 12, 20, and Y	LT (7), NCCR (2), VP1 (2)
2010	Laude H.C. (PLoS Pathogens)	DIPS-PCR	N/A	4	chr6, 9, 11, 15	LT (3), VP1 (1)
2011	Duncavage E.J. (J Mol Diagn)	MCPyV-specific Hybrid Capture NGS	SLOPE and Breakdancer	4	chr6, 8, 9	Not reported
2012	Martel-Jantin C. (Virology)	DIPS-PCR	N/A	19	chr1 (3),3 (1),4 (3), 5 (4), 6. (2), 14 (1), 18 (1) and 19 (2)	LT (13), ST (1), VP1(3), VP2 (2)
2013	Guastafierro A. (J Virol Methods)	Lambda phage library, 3’RACE, 5’RACE and Southern hybridization Library	N/A	4	chr3, 5, 6, 10	LT (3), ST (1)
2015	Doolittle-Hall et al., Cancers	N/A (Meta-analysis of published reports)	Meta-analysis of published reports	37	Multiple, Non-recurrent	Not reported
2017	Starrett G.J. (mBio)	Whole Genome NGS	Custom Pipeline	4	chr1, 6, 19, 20	LT (2), NCCR (2)
2019	Schrama D. et al., (Int J Cancer)	DIPS-PCR	N/A	5	chrs 1,2,6,11,12	LT (2), VP1 (4), VP2 (3)
2020	Slevin M.K. (J Mol Diagn)	Hybrid-capture and Massively parallel sequencing (Viropanel)	SvABA (structural variation and insertion/deletion analysis by assembly)	33	chr1 (5), 2 (2), 3 (1), 5 (7), 6 (4), 7 (2), 8 (2), 9 (2), 10 (1), 11 (4), 15 (1) and 16 (2)	LT (5), ST (2), VP1 (3), VP2 (1) and ** (1)
2020 (bioRxiv)	Czech-Sioli M. (bioRxiv)	Long-read third generation NGS, Nanochannel and Nanopore sequencing	Multiple	9	chr3, 4, 5 (4), 9, 11,13,20	LT (5), NCCR (2), VP1 (8), VP2 (4) and ** (1)
2020 (bioRxiv)	Garcia-Mulero S. (bioRxiv)	Human coding region sequencing NGS (Hybrid Capture)	No specialized analysis tool	1	chr 19	Not reported
2020	Starrett G.J. (Genome Med)	Hybrid-capture and Massively parallel sequencing (Viropanel)	Custom pipeline (oncovirus tools- suite - (https://github.com/gstarrett/oncovirus_tools))	25	chr1 (4), 2 (2), 3 (1), 5 (4), 6 (3), 7 (2), 8 (2), 9 (3), 10 (1), 11 (1), 15 (1), 16 (2) and 18 (1)	Not reported
2020	Arora R. (current study)	whole exome (Hybrid Capture)	HPVDetector	5	chr4, 7, 12, 16 and 17	LT, NCCR (2), VP1 (2)

** This was identified in the region between MCV LT and VP1 coding regions in the MCV genome.

**Table 2 viruses-12-00966-t002:** Summary of host and viral integration sites.

Breakpoints in the Human Genome		
Chromosome	Number of Sites	Genes Associated
1	14	PLA2G4A, RABGAP1L, MCOLN2, CASZ1 (I#1)
2	6	SF3B1 (I#1)
3	7	PTPRG (I#1), ATP11B, EEFSEC, ADAMTS9 (I)
4	6	SRD5A2L2, KCNIP4, RELL1, GRID2, FRG1(I),
5	21	TLX3, XRCC4, ISL1, PRLR, CAMK2A(I), CD74(I)
6	15	IL20RA, GMDS, MAP3K7IP2, CDKAL1 (I#4), MRDS1 (I#6), CDKAL1, RP3348–I23.2
7	5	
8	6	MYC
9	8	DENND1A;mir601, INPP5E (I)
10	3	
11	8	PARVA;TEAD1, LUZP2 (I#10;Tr#1), TTC9C (E#1), AHNAK (I)
12	3	AX747640, BEST3 (I#4;Tr#1), KMT2D (I#34)
13	1	DACH1 (I)
14	1	NOVA1
15	3	ATPBD4;XP_002343377
16	5	
17	1	DCAF7 (I#3)
18	2	CDH2
19	4	EVI5L, ECH1, GRWD1
20	3	SNTA1, CBFA2T2, BCL2L1 (E), TPX2 (I)
Y	1	
Total	123	
**Breakpoints in the Viral Genome**		
**Genomic Region**	**Number of Sites**	**Gene**
Large T Antigen	42	LT
Small T Antigen	4	ST
NCCR	8	
Viral Capsid Protein 1	23	VP1
Viral Capsid Protein 2	10	VP2
Between LT and VP1	2	
Total	89	

In brackets: I stands for Intron, Tr for Transcript, and E for Exon. The number following # is the corresponding Intron/Transcript/Exon number of the gene.

**Table 3 viruses-12-00966-t003:** Chimeric Sequences and Novel Integration Breakpoints from Current Study.

			MCPyV Genome		
Genome Location at Breakpoint	Gene	Forward Read Sequence	Genome Location at Breakpoint	Gene	Reverse Read Sequence
chr12_49429977	KMT2D (*I#34)	CACTTACTTCACTCGAAACAGTTTACTCCAGCCTAACCAATTCACTCACTGTTCTTGAAACATCCCATAGGTTTTCCCACCATTGCAG	MCV_DysKA_3082	Large T antigen	AGGGCCTATTAACAGTGGAAAAACAAGCTTTGCTGCAGCCTTAATAGATTTGCTAGAAGGGAAGGCCTTGAATATAAACTGTCCATCTGATAAACTACCT
chr17_61656843	DCAF7 (*I#3)	AAGAAGCCAAAATAAATTTTCTCAGATGTCACACTGGGCCATACAT CCTAGAACAAGGCTCAGTGGGAAGTAGGCTAGGCGTGGAATGTGGTTATTTATTCCTTTGTGTGGGTGAGCTTTCCGTGTGTATAGATCTTGTCTC	MCV_DysKA_172	NCCR	GAGGCCTCGGAGGCT AGGAGCCCCAAGCCTCTGCCAACTTGAAAAAAAAAAGTCACCTAGG
chr4_58486053	Intergenic	TAGATCCCTGAGGAATCGCCACACTGTCTTCCACAGTGGTTGAACTAGTATACAGTCT	MCV_DysKA_2201	VP1	TTATCTTTTCCTTCCATAGGTTGGCCTGACACTTTTGGCATTAAGTTGCTGAAGAGTGAGTTTGTTA
chr7_44780960	Intergenic	GTTGAAAAGGGCTGAGGAAAGAAAAATAGAGAACAAAAAGGGGGTGGGTCATTTTAAATTACTT	MCV_DysKA_175	NCCR	AGCCTCGGAGGCTAGGAGCCCCAAGCCTCTGCCAACTTGAAAAAAAAAAGTCACCTAGGC
chr16_85632533	Intergenic	TGGTTGGCACAAGAAACTGCCAAGCCACAGGCCCTGCTCCATACCCTCCTCTCTCCTACTCCCCCTCAATTATTTTTCTTGGATGTTGTGAATTAAATCTTCCAGTAACCGCAGGAGACGTCTTC	MCV_DysKA_1951	VP1	GTCCTTTTAAATGAGAATGGAGTGGGCCCTCTATGCAAAGGAGATGGCCTATTTATTAGCTGTGCAGA

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
