# Peer review of "Merkel Cell Polyomavirus in Merkel Cell Carcinoma: Integration Sites and Involvement of the KMT2D Tumor Suppressor Gene"

_viruses, 2020, doi:10.3390/v12090966_

Round 1
Reviewer 1 Report
This is a well-written and informative manuscript with original data and review of the literature. The work addressed an important issue in Merkel cell carcinoma biology: site of integration of the Merkel cell polyomavirus in virus-positive tumors.
Minor comments.
I suggest that the authors change the abbreviation MCV to MCPyV because the ICTV recommends using the abbreviation MCPyV for Merkel cell polyomavirus (MCV is used for the poxvirus Molluscum contagiosum virus, which was discoverd long before MCPyV).
Can the authors elaborate on the mechanism by which the lysine methyltransferase KMT2D contributes to cancer.
Table 1: supplement with viral load (number of viral genome copies) in case it was determined.
Author Response
We have included the relevant comments of the reviewer (in black) and our reply (in blue) both below.
I suggest that the authors change the abbreviation MCV to MCPyV because the ICTV recommends using the abbreviation MCPyV for Merkel cell polyomavirus (MCV is used for the poxvirus Molluscum contagiosum virus, which was discovered long before MCPyV).
Thank you for this correction. We have changed the abbreviation MCV to MCPyV in the manuscript.
Can the authors elaborate on the mechanism by which the lysine methyltransferase KMT2D contributes to cancer?
The following has been included in the paper (Discussion section):
Recent sequencing studies have shown that chromatin remodelling and histone modulators are frequently altered in cancers, with implications for tumour biology. KMT2D (lysine (K)-specific methyltransferase 2D), a.k.a. MLL2 (myeloid/lymphoid or mixed-lineage leukaemia 2, also known as ALR/MLL4), is one such histone methyltransferase that plays an important role in regulating gene transcription27. In particular, it targets lysine 4 of histone H3 (H3K4), whose methylations serve as a gene activation mark. KMT2D is essential for maintaining the level of global H3K4 monomethylation, with its enzymatic SET domain being directly responsible for this function. Furthermore, most of KMT2D binding sites are located in regions of potential enhancer elements. KMT2D has27 emerged as a frequently mutated gene in about 8.5% of all cancers28. KMT2D has been proposed to play either tumour suppressor or oncogenic roles, depending on cancer type. In MCC, KMT2D predominantly demonstrates inactivating mutations, supporting a tumour suppressor role29.
Table 1: supplement with viral load (number of viral genome copies) in case it was determined.
Thank you for this suggestion. The viral load was not determined or reported for all the publications listed and hence has not been included.
Reviewer 2 Report
The authors have adapted a pipeline designed to detect HPV sequences to identify Merkel cell polyomavirus (MCV) integrations in Merkel cell carcinomas (MCC). They analyze NGS data from seven MCCs and detect MCV integrations in five of them. The virus-host junctions were confirmed by PCR amplification and sequencing. In addition, they analyzed 91 previously published MCV integration sites.
A limited number of MCV integrations have been mapped and adding five more is useful. The following should be addressed:
1.The authors examined "7 whole exome sequencing datasets." (Line 82). Where did this NGS data come from? Did the authors sequence these tumors or is this from published data? The source and characteristics of the samples as well as a description of the sequencing data (what platform?, how deep?).
2.The authors have mapped virus-cell junctions. Each integration requires at least two virus-host junctions. This study would be more impactful if the structure of the entire integration event for each of the five samples was shown.
3.With this limited number of samples, I am not convinced that there is sufficient statistical power to support the enrichment of genes with LEF1 and ZEB1 binding sites.
Author Response
We have included the relevant comments of the reviewer (in black) and our reply (in blue) both below.
A limited number of MCV integrations have been mapped and adding five more is useful. The following should be addressed:
1.The authors examined "7 whole exome sequencing datasets." (Line 82). Where did this NGS data come from? Did the authors sequence these tumors or is this from published data? The source and characteristics of the samples as well as a description of the sequencing data (what platform?, how deep?).
We thank the reviewer for highlighting this. We have added the following relevant information to the manuscript.
Results section, first sentence now reads:
Of the 7 MCC whole exome sequencing datasets, previously published by Harms et. al7 , we predicted viral-host junctions in 5 (including 1 primary tumour and 4 metastases).
In Methods Section we have now included the following paragraph:
MCC whole-exome datasets were taken from the previously published study by Harms et al., Cancer Research 2015)7. The sequencing approach used by that study is briefly summarized here. Exome libraries were generated using Illumina TruSeq DNA Sample Prep Kit. Paired-end libraries were sequenced with the Illumina HiSeq 2000. Reads that passed the Illumina BaseCall software chastity filter were used for subsequent analysis. Sequence and alignment quality was assessed by FastQC and the Picard package, respectively. More details can be found in the supplement of Harms et al., Cancer Research 20157.
1.The authors have mapped virus-cell junctions. Each integration requires at least two virus-host junctions. This study would be more impactful if the structure of the entire integration event for each of the five samples was shown.
Yes, we agree with the reviewer that the study would be more impactful if we could map at least two of the viral-host junctions. However, this is a limitation of our experimental method/dataset and bioinformatically we could detect only single reads per dataset. We were also limited with the amount of genetic material for the MCC samples to experimentally determine both junctions. As described in the Limitations section:
In addition, our approach relies upon identification of off-target capture or shoulder reads of viral sequences within human whole exome sequencing data, thus generating relatively less sequencing depth than a sequencing approach specifically targeting MCPyV.
1.With this limited number of samples, I am not convinced that there is sufficient statistical power to support the enrichment of genes with LEF1 and ZEB1 binding sites.
Although relatively small, we believe that this sample set of 123 cases appears sufficiently powered for this unsupervised analysis. Our cohort is larger than a previously published cohort analysed by a similar approach (Doolittle-Hall), hence there is literature support for using a cohort of this size. Furthermore, because false negative results are much more likely than false positive results when an underpowered cohort is analysed with appropriate statistical rigor, our ability to detect statistically significant differences implies that our sample set is of adequate size for this purpose. Finally, we emphasize that the results of this analysis raise speculative hypotheses, regarding potential implications for open chromatin requirement and active oncogenic pathways in MCC, that require further investigation to confirm biological significance.
From our paper :
“Furthermore, the integration junctions in the human genome were observed to be enriched in genes having motifs/binding sites of known oncogenic transcription factors such as LEF1 27, MYB 28 and AREB6 (ZEB1) 29 along with other pathways (Supplementary Table 2); we speculate that this integration pattern might be related to an increased probability of open chromatin for these gene sets.”
Reviewer 3 Report
In the present study of “Merkel cell polyomavirus in Merkel cell carcinoma: Integration sites and involvement of the KMT2D tumor suppressor gene”, Arora and colleagues studied and reviewed the integration sites of Merkel cell polyomavirus (MCPyV) in Merkel cell carcinoma (MCC) samples. They showed and confirmed previous findings that no highly recurrent specific integration sites were identified in the genome of MCC samples and showed that chromosome 5 is most frequently integrated by MCPyV. They also found, in one case, the integration of MCPyV in the tumor suppressor gene KMT2D. The study expands the knowledge of the MCPyV integration in the host genome of MCC samples.
The main comment to the authors is only 5 of the 7 MCC exome sequencing were identified MCPyV integration sites, how does this affect the interpretation of the function of MCPyV integration in MCC development in MCPyV positive MCC? The authors should discuss this point.
Minor comment to the authors.
- I personally think MCPyV is more widely used for abbreviation of Merkel cell polyomavirus.
- Do you think the MCC will produce functional large T antigen for the four cases which integration sites were located in the NCCR region and VP1 region? This point is also intrigued to discuss.
Author Response
We have included the relevant comments of the reviewer (in black) and our reply (in blue) both below.
The main comment to the authors is only 5 of the 7 MCC exome sequencing were identified MCPyV integration sites, how does this affect the interpretation of the function of MCPyV integration in MCC development in MCPyV positive MCC? The authors should discuss this point.
It remains unclear whether tumors without detectable integration sites represent non-integrated virus (infectious/episomal) or technical artefact. We have added the following as a final sentence to “Limitations”:
Also, we cannot exclude the possibility that the lack of detected integration sites in 2/7 tumors may represent a technical limitation of our approach rather than true absence of viral integration.
Minor comment to the authors.
- I personally think MCPyV is more widely used for abbreviation of Merkel cell polyomavirus.
Yes. Thank you. This has been corrected.
- Do you think the MCC will produce functional large T antigen for the four cases which integration sites were located in the NCCR region and VP1 region? This point is also intrigued to discuss.
To address the reviewer’s concern, the following has been added to the Discussion section.
Several cases showed integration sites in the NCCR region (of these, our study reports 2 of the total 8), VP1 (our study reports 2 of the total 23) and VP2 regions (total 10 reported). The integration junctions may be disruptive or non-disruptive of gene expression and function, depending on the frame of fused product, mechanism of the integration, number of viral copies integrated etc. Based upon the previous description of our cases, 6 of 7 tumors had detectable LT protein expression by immunohistochemistry, and 6 of 6 tumors with RNASeq data had detectable LT transcript expression (Harms et al 2015, Supplemental Material)7. Therefore, all integration sites were permissive for LT expression, even those involving the NCCR. Of note, our approach is not designed to evaluate effects of integration on LT (especially relevant for integration sites that truncate the protein).